# HIRL-GAN: Hierarchical Mask-Guided Inpainting via GAN and Reinforcement Learning for Urban Occlusion Removal

## Abstract

In urban 3D reconstruction tasks, occlusions in architectural images often result in missing or distorted structures during point cloud reconstruction from images, thereby significantly affecting the accuracy of the final reconstruction. To address this issue, we propose HIRL-GAN, a **HI**erarchical and progressive inpainting framework that synergizes **R**einforcement **L**earning with **GAN**s, specifically designed for structured reconstruction of occluded building images. The proposed framework incorporates three key components: a hierarchical mask decomposition strategy that partitions complex occlusions into smaller sub-regions and restores them progressively to enhance structural stability; a reinforcement learning-based policy optimization mechanism that dynamically guides the reconstruction process at the sub-region level to improve restoration quality; and a self-attention-enhanced generator network that jointly models global semantics and local textures. In addition, we introduce a soft-mask guided training scheme to ensure smooth transitions and natural texture blending between restored and original regions. Extensive experiments on multiple image inpainting benchmarks demonstrate that HIRL-GAN achieves superior performance in structural recovery, detail reconstruction, and perceptual quality compared to existing state-of-the-art methods. Furthermore, ablation studies validate the effectiveness and robustness of the proposed RL-driven progressive mask restoration strategy under complex occlusion scenarios.

## 1 Introduction

In urban 3D reconstruction tasks Kerbl et al. (2023), architectural images are often occluded by foreground objects Zou & Hoiem (2020), leading to missing structures or geometric deviations in the reconstructed point clouds derived from these images. Such defects significantly undermine the accuracy and completeness of the final reconstruction results. As a crucial preprocessing step to improve reconstruction quality, recovering complete building structures from occluded images remains a key challenge Fritzsche et al. (2022).

Image inpainting, which aims to fill missing regions with visually and semantically plausible content Bertalmio et al. (2000); Pathak et al. (2016), has achieved notable progress in general settings. However, when applied to architectural imagery characterized by regular structures, large-scale occlusions, and high semantic constraints, existing methods often fail to simultaneously preserve structural consistency and texture realism—thereby limiting their effectiveness in downstream tasks such as 3D reconstruction.

Traditional inpainting approaches Guillemot & Le Meur (2013); Efros & Leung (1999) typically rely on low-level feature propagation or exemplar-based texture transfer. Although effective for small-scale or repetitive patterns, these methods lack the semantic modeling capacity required for handling complex structures such as building facades, leading to issues like texture drifting and discontinuities at boundaries. In recent years, deep generative models—including Generative Adversarial Networks (GANs) Pathak et al. (2016), diffusion models Saharia et al. (2022), and autoencoders Yu et al. (2019)—have shown remarkable progress in this domain by learning semantic features in an end-to-end manner. However, most of these models adopt a one-shot inpainting strategy, completing

the entire missing region in a single forward pass. This design lacks the flexibility to adaptively adjust the inpainting process based on contextual variations, often resulting in blurry, distorted, or semantically inconsistent outputs under large occlusions Pan et al. (2024). Furthermore, when applied to structured scenarios such as architectural imagery, these models tend to produce artifacts or misaligned global layouts Gsaxner et al. (2024).

To address these issues, recent studies have explored multi-stage or progressive inpainting frameworks Chen et al. (2024a), typically involving a coarse prediction phase followed by a refinement stage. Nonetheless, such approaches often rely on fixed restoration orders and are highly sensitive to the quality of the initial prediction. Structural errors introduced in the early stage are difficult to rectify later. Although attention mechanisms Kiani et al. (2024) have been incorporated to enhance contextual modeling, the lack of an effective mechanism for inpainting order optimization remains a major limitation.

To overcome these challenges, we propose a novel progressive occlusion removal framework that synergizes deep generative modeling with reinforcement learning to improve the stability and visual quality of large-scale structural inpainting. Our approach adopts a step-by-step restoration strategy to dynamically optimize the recovery sequence and incorporates attention mechanisms to enhance structural understanding. Extensive experiments are conducted on the Places2 dataset Zhou et al. (2017), the Oxford Buildings dataset, and a custom occlusion benchmark. Results show that our method outperforms state-of-the-art approaches in terms of PSNR, SSIM, and LPIPS Zhang et al. (2018), while achieving superior global structure preservation and local texture consistency.

The main contributions of this work are summarized as follows:

- Dynamic Repair Strategy: We combine GAN and deep reinforcement learning to learn a state-aware inpainting policy that adaptively determines the repair sequence and strategy, improving robustness and structural consistency for large occlusions.

- Progressive Small-Mask Inpainting: Large occluded regions are hierarchically decomposed into smaller sub-regions, which are restored progressively under RL guidance, combining global scheduling with local refinement for better texture continuity and detail fidelity.

- Structure-aware Priority Evaluation: Sub-regions are ranked using a scoring function based on edge sharpness, shape complexity, and semantic consistency, guiding repair order and providing meaningful priors for RL-based scheduling.

## 2 RELATED WORK

### 2.1 SAMPLE-BASED TRADITIONAL METHODS

Early inpainting techniques primarily rely on low-level texture propagation and structural cues Ashikhmin (2001); Efros & Freeman (2023). Non-parametric sampling Efros & Leung (1999) synthesizes textures without explicit parametric models, while exemplar-based inpainting Antonio et al. (2004) integrates exemplar matching and priority terms to ensure structural continuity. PatchMatch Barnes et al. (2009) further accelerated patch correspondence via randomized search. Though effective in texture repair for small regions, these approaches often struggle with large occlusions or semantically complex content due to the absence of high-level understanding.

### 2.2 DEEP LEARNING-BASED METHODS WITH GANS

The success of Convolutional Neural Networks (CNNs) spurred their adoption in image inpainting Xie et al. (2012); Ren et al. (2015); Köhler et al. (2014). Partial Convolution Liu et al. (2018) and Gated Convolution Yu et al. (2019) improved modeling for irregular holes. With the rise of Generative Adversarial Networks (GANs) Goodfellow et al. (2020), methods such as Co-Modulated GANs Zhao et al. (2021), CM-GAN Zheng et al. (2022), and SHU Xu et al. (2023) have demonstrated high-resolution and structure-aware inpainting. Structure–texture disentanglement Yeh et al. (2024), frequency-guided models Ding et al. (2024), attention-based strategies Dong et al. (2024), and multi-stage refinement Asad et al. (2025); Momen-Tayefeh et al. (2024) further advanced performance. However, these approaches often assume randomly distributed or small-scale holes, limiting robustness under structured occlusions like those found in urban environments.

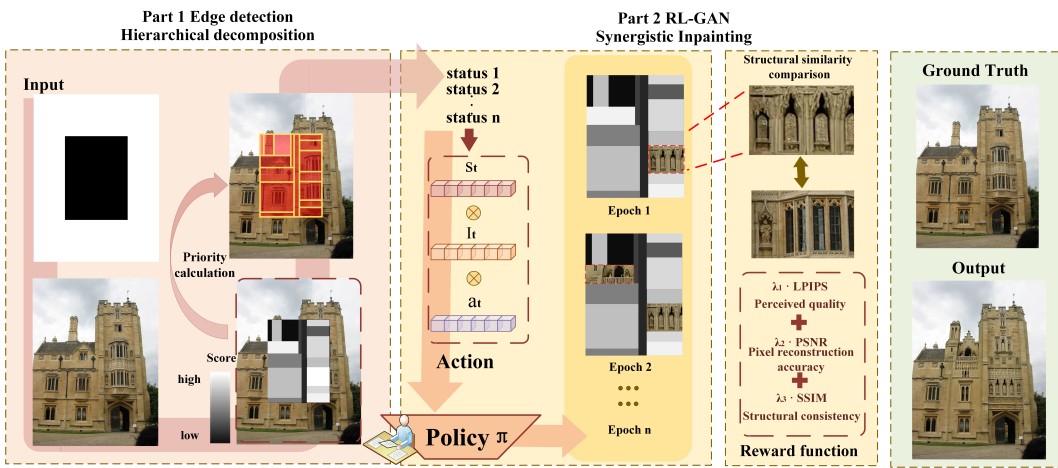

Figure 1: Reinforcement learning–guided hierarchical inpainting framework. The occluded region is first hierarchically decomposed based on surrounding edge cues and then ranked by a structure-aware priority function. The policy network $\pi$ observes the current state $s_t$, selects an action $a_t$ to restore a sub-region, and receives feedback from a multi-objective reward (SSIM, PSNR, LPIPS), progressively optimizing restoration order and quality.

## 2.3 DIFFUSION-BASED INPAINTING METHODS

Recently, diffusion models have emerged as powerful tools for generative tasks, including image inpainting. DDPM Ho et al. (2020) introduced a denoising-based sampling framework, further improved by DDIM Song et al. (2020) for faster inference. Palette Saharia et al. (2022) adapted diffusion for image-to-image translation, while RePaint Lugmayr et al. (2022) applied resampling-based diffusion for complex occlusions. Latent Diffusion Models (LDM) Rombach et al. (2022) reduced computational burden by operating in latent space. More recent approaches incorporate structure guidance Liu et al. (2024), mixed context modeling Xu et al. (2024), or semantic pre-inpainting Chen et al. (2024b). While effective in quality, these models are often computationally expensive and struggle with large-hole high-resolution inpainting under structural constraints.

To address these limitations, we propose a reinforcement learning–driven hierarchical inpainting strategy that combines the structural awareness of traditional methods, the generative capacity of GANs, and the adaptive scheduling capability of RL. Our framework progressively restores complex occlusions via region-level prioritization and patch-wise policy learning, significantly enhancing robustness in architectural scenarios.

# 3 METHODOLOGY

## 3.1 OVERALL FRAMEWORK

To address the challenges of large-scale occlusions and structural detail loss in architectural image inpainting, we propose HIRL-GAN, a progressive inpainting framework that combines hierarchical mask guidance with generative adversarial networks (GANs) and reinforcement learning. As illustrated in Figure 1, the framework comprises three key components: a hierarchical mask decomposition strategy, a GAN-based restoration module, and a reinforcement learning–driven decision-making system. These modules collaboratively partition the occluded region into semantically coherent sub-regions, progressively restore them with adaptive strategies, and ultimately produce high-fidelity reconstructions with enhanced structural and texture consistency.

## 3.2 HIERARCHICAL MASK DECOMPOSITION STRATEGY

To enhance the restoration quality of heavily occluded regions, we propose a novel hierarchical mask decomposition strategy, as illustrated in Figure 2. The goal is to partition the overall occluded

area into fine-grained sub-regions and prioritize their restoration in a structurally aware manner. This strategy operates in two stages: dynamic region partitioning, and regional priority assessment. These two stages jointly ensure structural consistency while improving both efficiency and robustness in restoring complex occlusions.

For the occluded regions in the input image $M_{\text{raw}}$, we design an edge-guided segmentation approach to perform structural partitioning. Specifically, we apply the Canny edge detection algorithm to extract prominent edge features $E = \text{Canny}(I)$ from the visible regions of the image, serving as structural cues to guide the decomposition process.

### 3.2.1 DYNAMIC REGION PARTITIONING

To enhance boundary constraints, we perform a morphological dilation operation on the extracted edge map $E$. This dilated edge map is then combined with the occlusion mask via a logical and operation, resulting in an edge-enhanced occlusion map $M_{\text{edge}}$.

$$M_{\text{edge}} = M_{\text{raw}} \wedge \text{Dilate}(E) \tag{1}$$

Subsequently, connected component analysis is applied to the edge-enhanced occlusion map $M_{\text{edge}}$, segmenting the original occluded region $M$ into a set of structurally independent and non-overlapping sub-regions $\{M_1, M_2, ..., M_n\}$ where $M_i \cap M_j = \emptyset$ for $i \neq j$.

This strategy ensures that each sub-region maintains structural completeness, enabling more effective local inpainting.

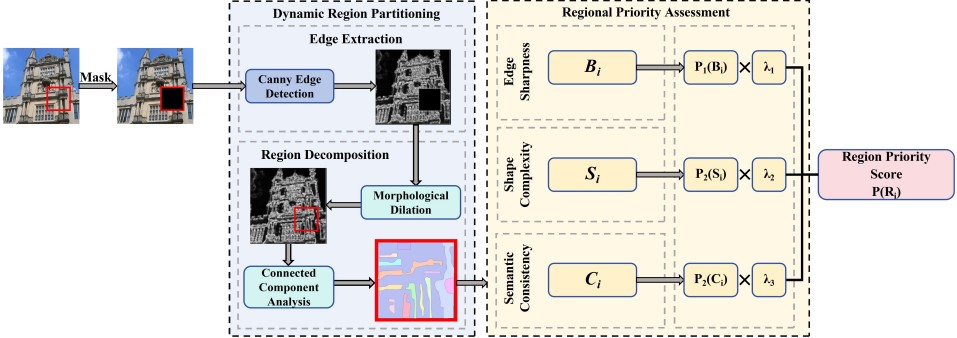

Figure 2: The hierarchical mask decomposition strategy consists of dynamic region partitioning and region priority assessment, where occluded areas are divided into semantically coherent sub-regions and ranked according to structural and contextual cues to guide progressive inpainting.

### 3.2.2 REGION PRIORITY ASSESSMENT

To determine the optimal restoration sequence of sub-regions, we design a priority evaluation function $P(M_i)$ that integrates structural, geometric, and semantic features. The core of this function is composed of the following three criteria.

Edge Sharpness $B_i$: Measures the sharpness of a region $M_i$'s boundary using gradient magnitude. A lower gradient typically indicates blurred or weak edges, which are often associated with broken textures and incomplete structures. Such regions are considered more challenging to reconstruct and are therefore assigned lower priority in the restoration sequence:

$$B_i = 1 - \frac{1}{|P_i|} \sum_{p \in P_i} \|\nabla I(p)\| \tag{2}$$

where $P_i$ denotes the set of pixels within the edge neighborhood of the sub-region $M_i$, and $\nabla I(p)$ represents the gradient magnitude at pixel $p$, computed using the Sobel operator.

Shape Complexity $S_i$: Quantifies the geometric irregularity of a region based on its compactness. Regions with more irregular shapes tend to have a higher perimeter-to-area ratio, indicating greater

structural complexity and potentially higher restoration difficulty. The shape complexity is defined as:

$$S_i = \frac{\text{Perimeter}(M_i)}{\text{Area}(M_i)} \tag{3}$$

Semantic Consistency $C_i$: Estimates the difficulty of semantic integration by computing the perceptual feature residual between the sub-region $M_i$ and its surrounding context. Deep features are extracted using a pretrained VGG19 network. The semantic inconsistency score is defined as:

$$C_i = \|f_{\text{context}} - f_{\text{region}}\|_1 \tag{4}$$

Taking all these factors into account, the final regional priority scoring function is defined as:

$$P(M_i) = \lambda_1 \cdot P_1(B_i) + \lambda_2 \cdot P_2(S_i) + \lambda_3 \cdot P_3(C_i) \tag{5}$$

where $\lambda_1$, $\lambda_2$ and $\lambda_3$ are hyperparameters, $P_1(B_i)$ is the structure-sensitive score, $P_2(S_i)$ is the shape complexity score, and $P_3(C_i)$ is the neighborhood context score, used to control the relative importance of different dimensions.

### 3.2.3 SCHEDULING STRATEGY AND HIERARCHICAL INPAINTING

Sub-regions are ranked by priority scores $P(M_i)$ and sequentially restored by the generator. A Progressive Sampling mechanism favors structurally complex areas during training, guiding the model to learn representative occlusion patterns early. This hierarchical scheduling enhances the model's adaptability to complex structures and supports subsequent reinforcement learning–based optimization.

### 3.3 ADVERSARIAL INPAINTING FRAMEWORK

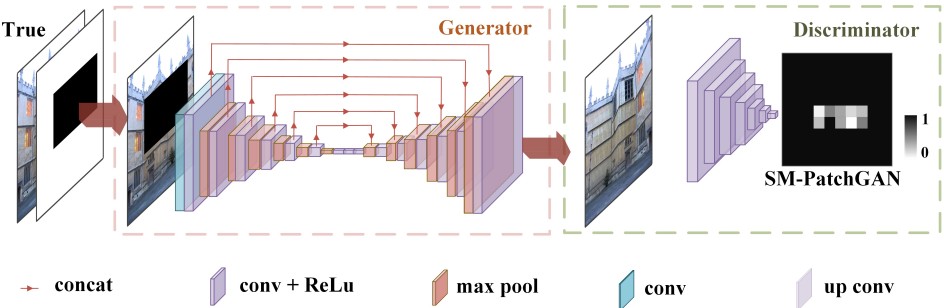

Figure 3: The GAN-based inpainting framework comprises an AOT-attentive generator and a soft-mask discriminator, enabling the model to synthesize realistic textures while maintaining structural consistency.

To enhance fine-grained texture and edge reconstruction, we adopt a GAN-based architecture integrating a U-Net-like generator, multi-scale AOT attention modules, and a Soft-Mask-guided discriminator (SM-PatchGAN), as illustrated in Figure 3.

### 3.3.1 GENERATOR NETWORK

The generator $G$ follows an encoder–decoder structure with skip connections. Inspired by AOT-GAN, we introduce its attention mechanism into our generator via stacked AOT modules, which facilitate structural reasoning through multi-scale aggregation and residual attention. The restored image $\hat{I}$ is synthesized from the occluded input $I$ and its mask $M$ via the generator: $\hat{I} = G(I, M)$.

### 3.3.2 LOSS FUNCTIONS

The discriminator adopts SM-PatchGAN, which extends the standard PatchGAN by introducing soft mask weighting, enabling region-aware evaluation of inpainted areas. The discriminator loss is

guided by a blurred soft mask $S$:

$$\mathcal{L}_D^{SM} = \sum_p S_p \cdot \log D_p(\hat{I}) \tag{6}$$

The generator is trained using a composite loss to balance visual realism, structural accuracy, and perceptual quality, including: $\mathcal{L}_{\text{adv}}$: adversarial loss, encouraging realistic generation; $\mathcal{L}_{\text{rec}}$: pixelwise reconstruction loss for content fidelity; $\mathcal{L}_{\text{perc}}$: perceptual loss based on VGG19 features to preserve semantic structure; $\mathcal{L}_{\text{style}}$: style loss based on Gram matrices to enhance texture consistency.

The final generator objective is a weighted sum of these terms:

$$\mathcal{L}_G = \lambda_{\text{adv}}\mathcal{L}_{\text{adv}} + \lambda_{\text{rec}}\mathcal{L}_{\text{rec}} + \lambda_{\text{perc}}\mathcal{L}_{\text{perc}} + \lambda_{\text{style}}\mathcal{L}_{\text{style}} \tag{7}$$

where $\lambda_{\text{adv}}, \lambda_{\text{rec}}, \lambda_{\text{perc}}, \lambda_{\text{style}}$ are scalar weights controlling each term's contribution.

### 3.4 REINFORCEMENT LEARNING DECISION MODULE

To enhance the scheduling capability of local reconstruction strategies for occluded regions, we introduces a reinforcement learning module based on Deep Deterministic Policy Gradient (DDPG). This module guides the inpainting process to perform local inference in a more reasonable order, thereby improving the overall structural consistency and restoration quality. The entire process iteratively optimizes the inpainting policy $\pi$, enabling a complete workflow of progressive patchwise restoration→synthesis→ refinement.

#### 3.4.1 REINFORCEMENT LEARNING FORMULATION

To guide the hierarchical restoration of occluded regions, we model the inpainting process as a reinforcement learning (RL) problem. Given an image $I$ and its occlusion mask $M$, we first decompose $M$ into sub-regions $\{M_i\}_{i=1}^N$, which are treated as individual restoration units. The RL agent progressively restores each sub-region, producing an intermediate image $\tilde{I}$, followed by a final global refinement to obtain the completed output $\hat{I}$.

We define the state space $s_t = (I_t, M_t, (i, j))$, encoding the current partially inpainted image $I_t$, the corresponding mask $M_t$, and the spatial coordinates $(i, j)$ of the active patch. The action space $a_t = \pi_\theta(s_t) = G_\theta(I_t, M_t, (i, j))$ corresponds to the content predicted for the current sub-region by the inpainting policy $\pi_\theta$.

The reward function evaluates restoration quality by combining structural, pixel-level, and perceptual metrics:

$$r_t = \lambda_1 \text{SSIM}(a_t, y_{ij}) + \lambda_2 \text{PSNR}(a_t, y_{ij}) - \lambda_3 \text{LPIPS}(a_t, y_{ij}), \tag{8}$$

where $y_{ij}$ denotes the ground-truth patch, and the weights $\lambda_1, \lambda_2, \lambda_3$ balance the contributions of each metric.

#### 3.4.2 DDPG POLICY UPDATE AND OU NOISE EXPLORATION

To enable policy learning in continuous action spaces, we adopt the Deep Deterministic Policy Gradient framework for training:
Critic Network Loss:

$$\mathcal{L}_{\text{Critic}} = \left(Q(s_t, a_t) - [r_t + \gamma Q(s_{t+1}, a_{t+1})]\right)^2 \tag{9}$$

Actor Network Optimization Objective:

$$\mathcal{L}_{\text{Actor}} = -Q(s_t, \pi(s_t)) \tag{10}$$

To mitigate slow policy convergence and action oscillations in the early training stages, we incorporate temporally correlated Ornstein-Uhlenbeck (OU) noise into the action outputs. This strategy enhances exploration by introducing smooth, mean-reverting perturbations, with the noise intensity gradually decaying as training progresses.

A replay buffer mechanism is employed to store quintuple tuples $(s_t, a_t, r_t, s_{t+1}, d_t)$, facilitating sampling-based updates and enhancing training stability.

### 3.4.3 POLICY EXECUTION AND REFINEMENT STAGE

The restoration policy $\pi_\theta$ generated by the reinforcement learning module, guides both the repair order and method for each small occluded region. Once all local areas have been restored, they are stitched together to form an intermediate image $\tilde{I}_t$. Finally, the trained generator performs a full-image refinement to enhance global details, resulting in the final restored output $\hat{I}$.

## 4 EXPERIMENTS

### 4.1 DATASETS

To support our proposed HIRL-GAN framework, we design a three-stage data pipeline. First, we pretrain the GAN inpainting network on the large-scale Places2 dataset, which provides diverse scene-level supervision to enhance global structure and texture modeling. Then, we finetune the GAN jointly with the reinforcement learning module using the Oxford Buildings Dataset, which offers high-resolution architectural images more representative of our target domain.

To evaluate occlusion removal performance in realistic scenarios, we construct a building-specific test set by compositing object masks from COCO Lin et al. (2014) onto Oxford building images, as illustrated in Figure 5 (Input). This synthetic occlusion dataset contains diverse occluder shapes, scales, and positions, simulating challenging real-world cases. All datasets are split into 80% for training and 20% for testing.

### 4.2 EVALUATION METRICS

To comprehensively evaluate the performance of our progressive inpainting method, we adopt PSNR, SSIM, and LPIPS as evaluation metrics, measuring pixel accuracy, structural similarity, and perceptual quality respectively. We conduct comparative experiments against state-of-the-art methods including LaMa Suvorov et al. (2022), AOT-GAN Zeng et al. (2022), MI-GAN Sargsyan et al. (2023), MAT Li et al. (2022),ir-sde Luo et al. (2023) and StrDiffusion Liu et al. (2024) under identical test settings. The results demonstrate that our method achieves superior performance across all metrics, particularly in scenarios with large-scale or unstructured occlusions, where it better preserves global structure and local details.

### 4.3 RESULTS

### 4.3.1 QUANTITATIVE EVALUATION

We evaluate the proposed HIRL-GAN on a composite dataset built from COCO and Oxford Buildings, comparing it with state-of-the-art inpainting approaches including LaMa, AOT-GAN, MI-GAN, MAT, ir-sde and StrDiffusion. The comparison is conducted under identical test settings using the evaluation metrics introduced in Evaluation Metrics.

Table 1: Comparison between the proposed method and existing approaches.

| Method | PSNR↑ | SSIM↑ | LPIPS↓ |
|---|---|---|---|
| AOT-GAN | 31.76 | 0.8193 | 0.0428 |
| MI-GAN | 36.71 | 0.9485 | 0.0418 |
| Lama | 36.82 | 0.9302 | 0.0389 |
| MAT | 35.41 | 0.8989 | 0.0428 |
| ir-sde | 36.06 | 0.9237 | 0.0423 |
| StrDiffusion | 34.50 | 0.8850 | 0.0425 |
| **HIRL-GAN(Ours)** | **37.68** | **0.9511** | **0.0328** |

As shown in Table 1, our method achieves the best PSNR and SSIM, indicating improved reconstruction accuracy and structural consistency. It also yields the lowest LPIPS, suggesting better perceptual quality.

To further investigate the impact of occlusion ratio on inpainting performance, we evaluate each method under three different occlusion levels: 0–20%, 20–40%, and 40–60%. As shown in Table 2, the reconstruction quality of all methods degrades as the occlusion ratio increases. However, our model consistently maintains superior performance, particularly under high occlusion rates. These

Table 2: Inpainting performance under varying occlusion ratios.

| Method | 0%–20% | | | 20%–40% | | | 40%–60% | | |
| --- | --- | --- | --- | --- | --- | --- | --- | --- | --- |
| | PSNR↑ | SSIM↑ | LPIPS↓ | PSNR↑ | SSIM↑ | LPIPS↓ | PSNR↑ | SSIM↑ | LPIPS↓ |
| AOT-GAN | 31.75 | 0.819 | 0.041 | 31.72 | 0.7922 | 0.093 | 31.37 | 0.7434 | 0.162 |
| MI-GAN | 36.69 | 0.948 | 0.015 | 35.11 | 0.8519 | 0.039 | 33.81 | 0.8218 | 0.097 |
| Lama | 36.82 | 0.931 | 0.014 | 35.89 | 0.8491 | 0.035 | 34.23 | 0.8427 | 0.098 |
| MAT | 36.51 | 0.942 | 0.019 | 34.99 | 0.8841 | 0.045 | 33.71 | 0.8153 | 0.088 |
| ir-sde | 36.55 | 0.945 | 0.016 | 34.95 | 0.8479 | 0.042 | 33.65 | 0.8183 | 0.098 |
| StrDiffusion | 34.20 | 0.895 | 0.040 | 33.00 | 0.8351 | 0.065 | 32.41 | 0.7802 | 0.120 |
| **HIRL-GAN(Ours)** | **37.67** | **0.949** | **0.013** | **36.21** | **0.8861** | **0.034** | **34.91** | **0.8531** | **0.069** |

results demonstrate the effectiveness of our progressive strategy in preserving structure and visual quality, especially under challenging large-occlusion conditions.

### 4.3.2 QUALITATIVE EVALUATION

To evaluate the generalization capability of our model under diverse occlusion patterns, we conduct qualitative comparisons on the Oxford Buildings dataset with free-form masks. As shown in Figure 4, HIRL-GAN demonstrates superior restoration quality compared to several representative inpainting baselines. It effectively maintains architectural structural consistency while generating more natural texture details, outperforming common failures such as geometric distortion or texture artifacts observed in other methods. This highlights the robustness and adaptability of our framework under complex and irregular occlusions. Furthermore, to assess the model's performance in

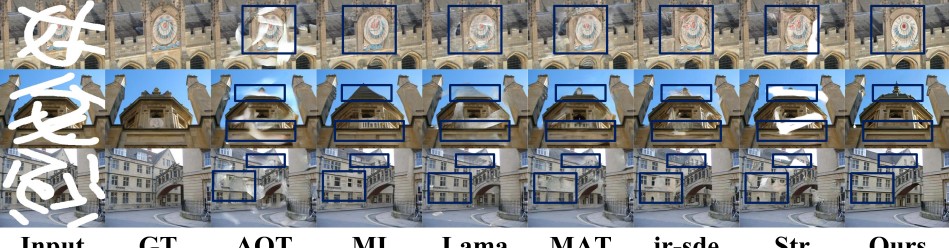

**Input      GT      AOT      MI      Lama      MAT      ir-sde      Str      Ours**

Figure 4: Comparison between the proposed method and existing approaches under random free-form occlusions on standard datasets. GT denotes Ground Truth.

removing foreground occluders from architectural scenes in real-world scenarios, we construct a composite dataset by overlaying COCO object masks onto Oxford Buildings images and evaluate the qualitative performance under different occlusion ratios (0–60%), as shown in Figure 5.

Under light occlusion (0–20%), most baseline methods manage to complete the missing content, but often result in blurry textures or inconsistent boundaries. In contrast, our HIRL-GAN yields clearer local details and better global coherence through progressive restoration. As the occlusion increases to 20–40%, existing methods tend to suffer from structural discontinuities and unnatural textures, whereas our method leverages reinforcement learning to prioritize critical sub-regions, achieving more semantically aligned results. In heavily occluded scenarios (40–60%), baseline methods exhibit noticeable artifacts and semantic collapse. In comparison, our method gradually accumulates reliable content and applies global refinement, effectively preserving structural integrity and ensuring perceptual quality of the restored output.

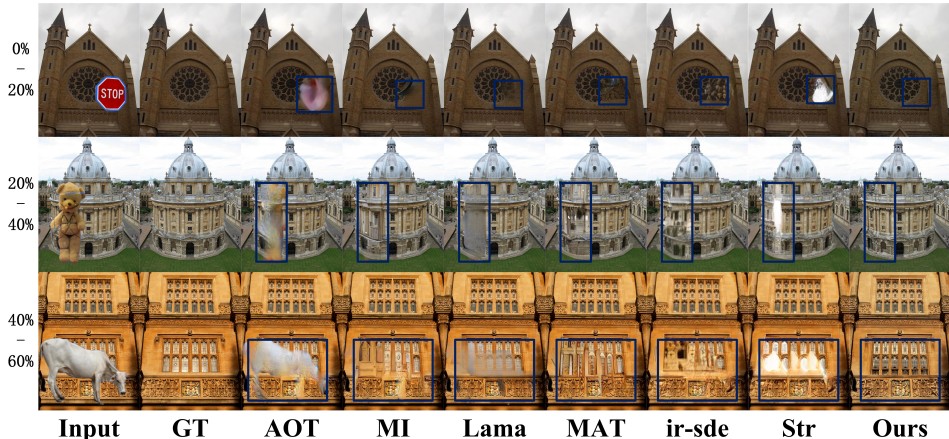

Figure 5: Comparison of inpainting results of different methods under different occlusion levels (0–20%, 20–40%, 40–60%).

## 4.4 ABLATION STUDY: ANALYSIS OF THE REINFORCEMENT LEARNING MODULE

To verify the practical contribution of the reinforcement learning (RL) module within the overall framework, this section designs a set of ablation experiments comparing the inpainting performance with and without RL guidance. The experiments are based on the complete HIRL-GAN framework, controlling variables only by the presence or absence of the reinforcement learning module, denoted as:

**HIRL-GAN w/ RL**: Incorporates the DDPG-based reinforcement learning module, repairing each sub-region sequentially according to the priority scheduling order.

**HIRL-GAN w/o RL**: Does not use reinforcement learning, performing repairs on all sub-blocks in a fixed order, with the generator architecture remaining unchanged.

Table 3 shows that RL improves all metrics, notably reducing LPIPS. As seen in Figure 6, the RL-guided variant produces more coherent structures and realistic textures, while the baseline often yields artifacts and inconsistent boundaries.

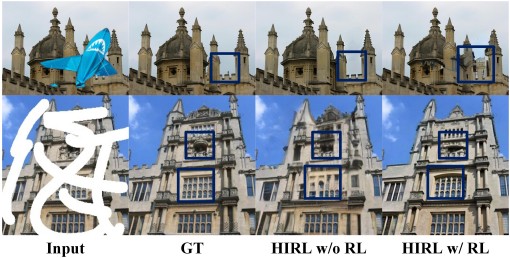

Table 3: Performance comparison of HIRL-GAN with and without reinforcement learning.

| Method | PSNR↑ | SSIM↑ | LPIPS↓ |
|---|---|---|---|
| HI RL-GAN w/o RL | 34.64 | 0.9317 | 0.0413 |
| **HIRL-GAN w/ RL** | **36.57** | **0.9472** | **0.0387** |

Figure 6: Visual comparison of HIRL-GAN with and without RL.

These results confirm that reinforcement learning plays a key role in adaptive inpainting scheduling, improving robustness and structural alignment in complex occlusion scenarios.

## 5 CONCLUSION

In this work, we propose HIRL-GAN, a hierarchical and reinforcement learning-guided framework for image inpainting. By decomposing large occlusions into subregions and leveraging a score-driven Actor-Critic strategy to optimize restoration order, our method achieves superior structure preservation and texture consistency. Extensive experiments demonstrate competitive performance under challenging occlusion scenarios.

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

# A    APPENDIX

## A.1    USE OF LARGE LANGUAGE MODELS

We acknowledge the use of a large language model (ChatGPT, developed by OpenAI) to assist in the language refinement of this manuscript. The model was used strictly for editing and improving the clarity, grammar, and style of human-written text. All scientific content, experimental design, results, and analysis were developed independently by the authors without the use of AI-generated content. The language model was not involved in generating novel ideas, code, equations, or experimental results.

## A.2    TRAINING ENVIRONMENT AND HARDWARE CONFIGURATION

All experiments were implemented using the PyTorch framework and conducted on a single NVIDIA RTX 3090 GPU (24GB memory). The GAN module was optimized using the Adam optimizer (learning rate = 0.0001, batch size = 8), with a soft-mask strategy employed to smooth mask boundaries and improve the naturalness of the inpainted regions.

For the reinforcement learning (RL) component, we adopt the Deep Deterministic Policy Gradient (DDPG) algorithm. The state space is defined by the masked image patch, its corresponding mask, and the positional index, while the action space outputs the restoration operation for each sub-region. The reward function integrates SSIM, PSNR, and LPIPS to guide the policy network in dynamically optimizing both the patch-wise restoration sequence and the generation quality.

## A.3    ADDITIONAL VISUALIZATIONS

To further illustrate the effectiveness of HIRL-GAN, we provide additional qualitative comparisons under three occlusion levels: 0–20%, 20–40%, and 40–60%. These supplementary visual results (Figures 7–9) compare HIRL-GAN with representative inpainting methods, including LaMa, AOT-GAN, MI-GAN, MAT, ir-sde and StrDiffusion. The comparisons demonstrate how the proposed progressive inpainting strategy adapts to varying occlusion ratios. Even under severe occlusions (40–60%), HIRL-GAN preserves structural consistency and texture fidelity, achieving restorations that are visually coherent and natural.

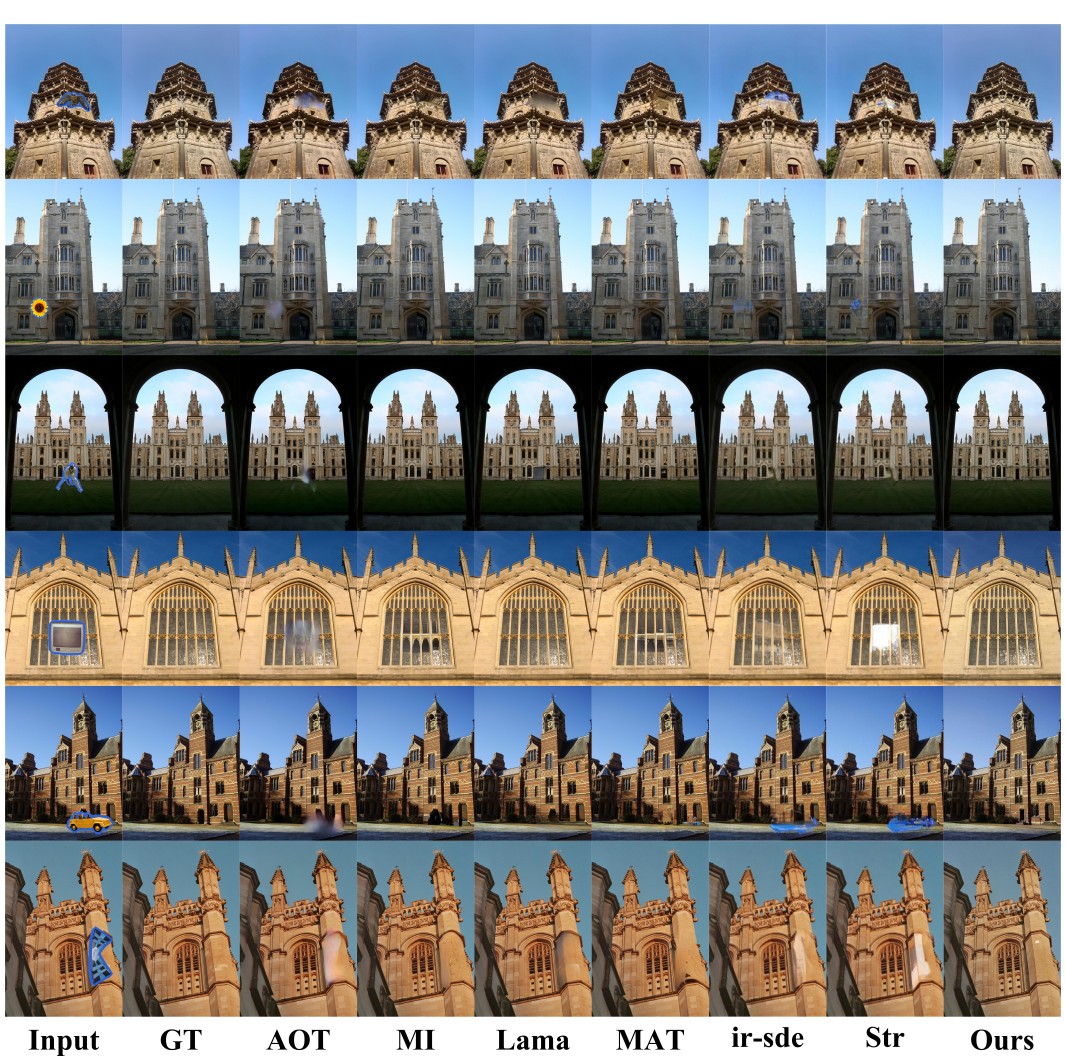

Figure 7: Comparison of different methods under 0–20% occlusion conditions.

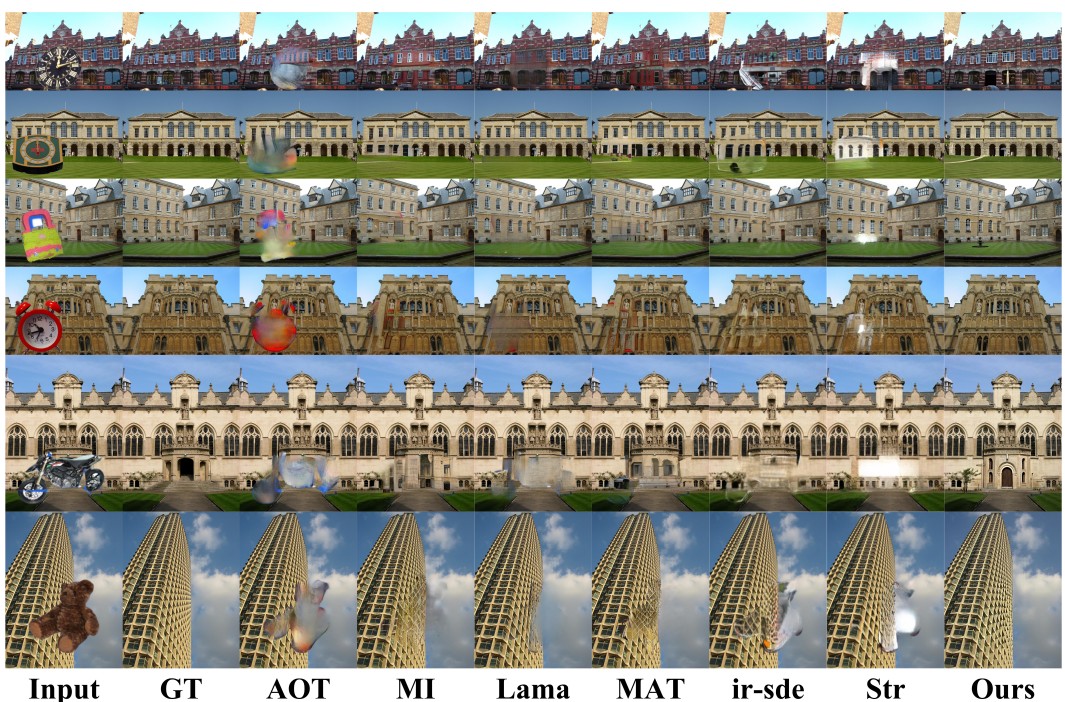

**Input    GT    AOT    MI    Lama    MAT    ir-sde    Str    Ours**

Figure 8: Comparison of different methods under 20–40% occlusion conditions.

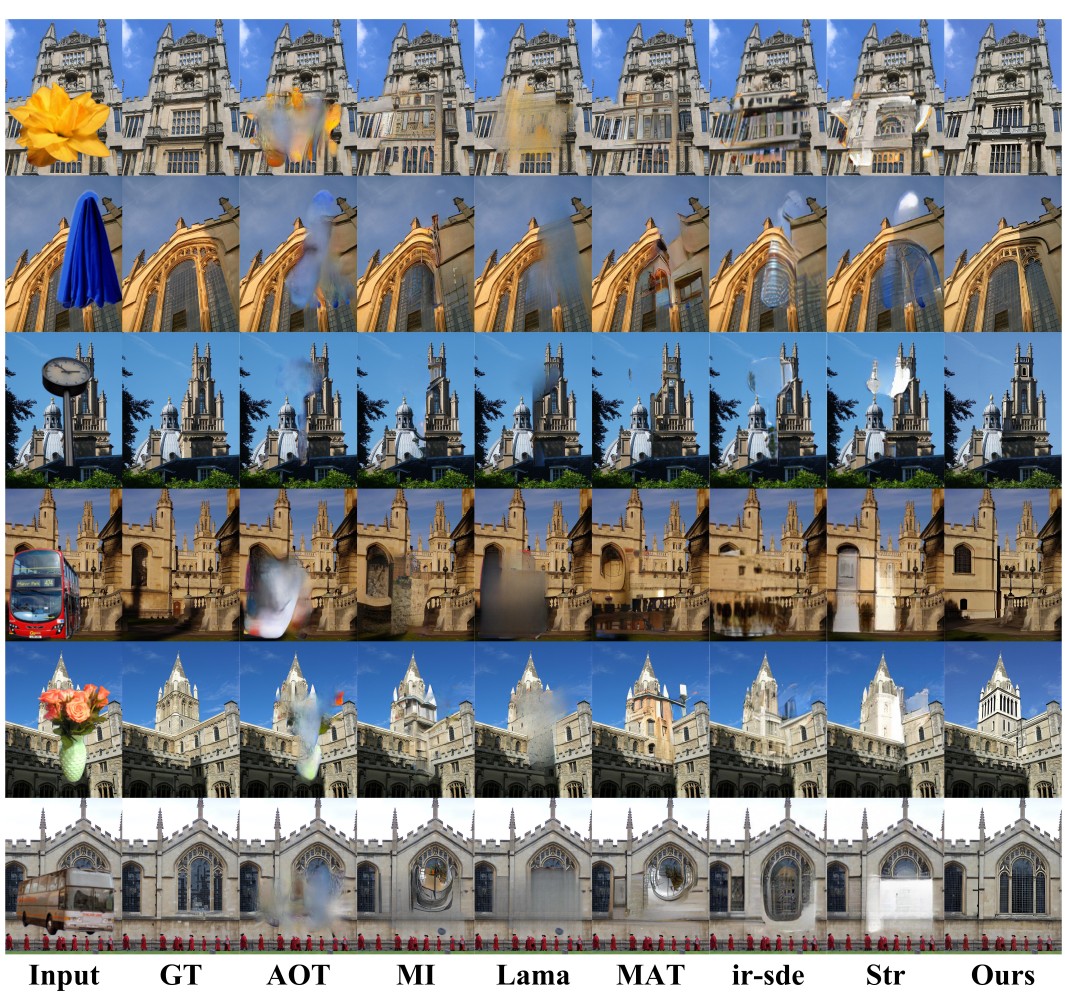

Figure 9: Comparison of different methods under 40–60% occlusion conditions.

