# OpenReview forum: "HIRL-GAN：Hierarchical Mask-Guided Inpainting via GAN and Reinforcement Learning for Urban Occlusion Removal"
_ICLR.cc/2026/Conference — ICLR 2026 Conference Withdrawn Submission_

### Official Review · Reviewer_zrFG · 2025-10-21

**Soundness:** 2
**Presentation:** 2
**Contribution:** 3
**Rating:** 4
**Confidence:** 4

**Summary:**

This paper proposes HIRL-GAN, a hierarchical and reinforcement learning (RL)-guided GAN framework for image inpainting in urban 3D reconstruction scenarios. The method decomposes large occluded regions into smaller sub-regions through a hierarchical mask decomposition strategy and employs a reinforcement learning policy (based on DDPG) to guide the restoration sequence dynamically. A self-attention-enhanced GAN further improves structural and texture consistency. Experiments on datasets such as Oxford Buildings and Places2 suggest that HIRL-GAN achieves competitive PSNR, SSIM, and LPIPS results compared with existing inpainting models like LaMa, AOT-GAN, and StrDiffusion.

**Strengths:**

（1）The hierarchical mask decomposition for large occlusions is a creative adaptation that could improve structure-aware inpainting, particularly in architectural imagery.
（2）The paper is generally well organized, with consistent formatting and logical flow across sections.
（3）Urban occlusion removal and structured inpainting could be potentially extended to other structured inpainting domains.

**Weaknesses:**

（1）The computational complexity of reinforcement learning is not discussed in depth. The introduction of reinforcement learning incurs greater computational overhead, and the paper does not provide a corresponding computational complexity analysis.

（2）This article only focuses on completing architectural scenes,  thereby somehow narrowing its impact.

（3） The RL component adds significant complexity but lacks detailed convergence analysis, hyperparameter sensitivity, or release of code/models.

（4）Missing comparisons with recent inpainting frameworks that have surpassed GAN-based models in perceptual quality.

（5）Figure legends are overly brief and fail to highlight critical visual differences, such as lacking zoom-in insets.

（6）Lack of analysis on failure cases.

**Questions:**

(1) What is the computational overhead of adding RL to GAN training compared to a standard GAN? I am concerned about the computational complexity of introducing reinforcement learning for 2D inpainting. Please provide the corresponding computational complexity analysis for each component.

(2)Can the method generalize beyond architectural images (e.g., natural occlusions, objects)?

(3) The iterative framework seems computationally expensive. For practical applications, this could be prohibitive. You can provide more detailed analysis of the speed-accuracy trade-offs.

(4) As I know, there are several recent works and released benchmarks. The authors should compare their results with these works or discuss them in the related work section if the code is not available:
[1] Wang Y, Cao C, Yu J, et al. Towards Enhanced Image Inpainting: Mitigating Unwanted Object Insertion and Preserving Color Consistency[C]//Proceedings of the Computer Vision and Pattern Recognition Conference. 2025: 23237-23248.
[2]Zhang L, Yu Y, Yao J, et al. High-Fidelity Image Inpainting with Multimodal Guided GAN Inversion[J]. International Journal of Computer Vision, 2025: 1-18.

(5) Please clarify whether the hierarchical mask partitioning introduces discontinuities when merging sub-regions and how these are resolved.

(6) Eq.8 ends with a comma, while Eq.7 does not. The author should maintain consistency in this formatting.

(7) Does the current method depend on pre-trained GAN model? If the backbone network is replaced (e.g., with diffusion model), will the performance change?

---

### Official Review · Reviewer_eRBH · 2025-10-27

**Soundness:** 3
**Presentation:** 3
**Contribution:** 3
**Rating:** 2
**Confidence:** 4

**Summary:**

This paper addresses image inpainting through an iterative process that divides the missing region into multiple sub-regions and fills them one by one. To create these sub-regions, the authors first extract Canny edges from the visible areas, then apply morphological dilation to extend the edges into the masked region. A connected component analysis is then used to identify several disjoint sub-regions. Each sub-region is assigned a priority score based on edge sharpness, shape complexity, and semantic consistency, which are estimated either mathematically or through pre-trained feature similarity. These hierarchical masks are used to train the inpainting model in a more fine-grained way, combined with a reinforcement learning algorithm that explicitly optimizes LPIPS, PSNR, and SSIM metrics.

**Strengths:**

The idea of decomposing the masked region into smaller sub-regions is interesting and creative.

**Weaknesses:**

1. The overall motivation is unclear, and the proposed mask decomposition method requires more discussion and ablation studies.
    - The morphological dilation step seems to work well only for highly structured objects, which may limit the generalizability of the method.
    - The paper claims that "A Progressive Sampling mechanism favors structurally complex areas during training, guiding the model to learn representative occlusion patterns early. This hierarchical scheduling enhances the model’s adaptability to complex structures and supports subsequent reinforcement learning–based optimization." However, this statement lacks experimental evidence. Further discussion and ablations are recommended.
2. The RL component is unconvincing because it directly optimizes the evaluation metrics of inpainting. Moreover, the experiments only report those same metrics used in RL training, which makes the results less reliable. It is recommended to include additional inpainting metrics, especially ones not used during training.
3. The abstract mentions urban 3D reconstruction, but the paper only presents experiments on the Oxford Building dataset, which is a 2D inpainting task. This narrow focus limits the generality of the proposed method.
4. The paper should discuss and compare its approach with more recent inpainting methods.
5. The experiments only test masks covering less than 60% of the image, which raises concerns about performance on large-hole inpainting tasks.
6. The experimental setup seems unrealistic, as the missing regions are filled using objects pasted from other images. This ignores lighting, shadows, and other contextual effects, making the task much easier than real-world inpainting.

**Questions:**

1. Why are the HIRL-GAN results in Table 1 and Table 3 different?
2. Why did the authors choose a GAN backbone, when the method appears more suitable for autoregressive or discrete diffusion models?

---

### Official Review · Reviewer_ENyU · 2025-11-02

**Soundness:** 2
**Presentation:** 3
**Contribution:** 2
**Rating:** 4
**Confidence:** 3

**Summary:**

This paper introduces HIRL-GAN, a Hierarchical and Reinforcement Learning–guided image inpainting framework designed for urban occlusion removal in architectural imagery. The motivation stems from the observation that existing GAN- or diffusion-based inpainting methods, while effective for small or random holes, struggle to maintain structural regularity and semantic consistency when handling large-scale occlusions in urban environments.

**Strengths:**

1. HIRL-GAN’s progressive, decision-aware restoration distinguishes it from traditional inpainting models that operate in a single forward pass.
2. The edge-guided dynamic region partitioning ensures that local patches are semantically and geometrically coherent.
3. Across multiple occlusion levels (0–60%), HIRL-GAN consistently outperforms baselines, especially at high occlusion ratios, where other models fail to maintain geometry

**Weaknesses:**

1. The RL objective is largely empirical (weighted PSNR/SSIM/LPIPS) rather than derived from a principled generative-theoretic framework.
2. While the Related Work section reviews diffusion-based inpainting, experiments do not include recent latent or control-based diffusion baselines (e.g., RePaint, ControlNet-LDM).
3. Some mathematical formulations (e.g., the priority scoring function Eq. 5, or the reward Eq. 8) are only verbally justified; a clearer explanation of hyperparameter tuning or sensitivity analysis would improve interpretability.
4. The approach is highly tailored to structured architectural imagery. It remains unclear whether the learned policy generalizes to natural scenes or non-rectilinear layouts.

**Questions:**

see the weaknesses

---

### Official Review · Reviewer_rBXf · 2025-11-04

**Soundness:** 3
**Presentation:** 3
**Contribution:** 3
**Rating:** 4
**Confidence:** 4

**Summary:**

This paper introduces HIRL-GAN, a new framework for image inpainting (occlusion removal) designed specifically for urban architectural images. The primary goal is to improve the accuracy of downstream 3D reconstruction tasks, which are often damaged by occlusions.





Instead of traditional single-pass methods, HIRL-GAN uses a hierarchical and progressive approach:



Mask Decomposition: It first breaks down large, complex occlusions into smaller, structurally-aware sub-regions.


RL-Guided Inpainting: It then uses Reinforcement Learning (RL) to create a policy that guides the restoration process for these sub-regions, aiming for better structural consistency.



GAN Generation: The actual image content is generated by a GAN equipped with self-attention and a special "soft-mask" guided discriminator.

Experiments on datasets like Oxford Buildings show that HIRL-GAN outperforms other state-of-the-art methods (e.g., LaMa, AOT-GAN) on standard 2D metrics (PSNR, SSIM, LPIPS), especially when dealing with very large occlusions (40-60% of the image).

**Strengths:**

Interesting idea: The core idea of using Reinforcement Learning to guide the inpainting sequence is highly novel. It shifts the problem from just how to fill a hole to in what order to fill it, which is crucial for structured objects like buildings.

**Weaknesses:**

Performance: I found that adding RL did not lead to significant improvement, according to the ablation study.

Critical Ambiguity in RL's Role: The paper is confusing about what the RL agent actually does. The introduction and abstract claim the RL optimizes the "repair sequence" or "repair order.  However, the method section (3.4.1) defines the patch location $(i, j)$ as part of the state (the input), and the generated content as the action (the output). This implies the RL agent does not choose the order; it only learns how to generate pixels for a patch it is given.

Missing 3D Reconstruction Validation:  The paper's entire motivation is to improve 3D reconstruction. However, all experiments are based on 2D image metrics (PSNR, LPIPS, etc.). There is no evidence that the images repaired by HIRL-GAN actually lead to better 3D point clouds or models than those from other baselines.

**Questions:**

See the weakness.

---

### Note · Authors · 2025-11-18

I have read and agree with the venue's withdrawal policy on behalf of myself and my co-authors.